# Amorphous Silicon Thin-Film Solar Cells on Fabrics as Large-Scale Detectors for Textile Personal Protective Equipment in Active Laser Safety [note 1]

**DOI:** 10.3390/ma16134841

**Published:** 2023-07-05

**Authors:** Annett Gawlik, Uwe Brückner, Gabriele Schmidl, Volker Wagner, Wolfgang Paa, Jonathan Plentz

**Affiliations:** Leibniz Institute of Photonic Technology (Leibniz IPHT), Department of Functional Interfaces, Albert-Einstein-Strasse 9, 07745 Jena, Germany; uwe.brueckner@leibniz-ipht.de (U.B.); gabriele.schmidl@leibniz-ipht.de (G.S.); wolfgang.paa@leibniz-ipht.de (W.P.)

**Keywords:** personal protective equipment (PPE), active laser safety, textile sensors, thin-film solar cells

## Abstract

Laser safety is starting to play an increasingly important role, especially when the laser is used as a tool. Passive laser safety systems quickly reach their limits and, in some cases, provide inadequate protection. To counteract this, various active systems have been developed. Flexible and especially textile-protective materials pose a special challenge. The market still lacks personal protective equipment (PPE) for active laser safety. Covering these materials with solar cells as large-area optical detectors offers a promising possibility. In this work, an active laser protection fabric with amorphous silicon solar cells is presented as a large-scale sensor for continuous wave and pulsed lasers (down to ns). First, the fabric and the solar cells were examined separately for irradiation behavior and damage. Laser irradiation was performed at wavelengths of 245, 355, 532, and 808 nm. The solar cell sensors were then applied directly to the laser protection fabric. The damage and destruction behavior of the active laser protection system was investigated. The results show that the basic safety function of the solar cell is still preserved when the locally damaged or destroyed area is irradiated again. A simple automatic shutdown system was used to demonstrate active laser protection within 50 ms.

## 1. Introduction

Today, lasers are used in countless applications. The laser is becoming an increasingly powerful tool with enormous power densities and pulse energies. As a result, the dangers posed by laser radiation are increasing significantly. Previous passive laser protection concepts, such as laser-proof materials, often no longer meet the requirements of laser protection because they cannot withstand the high power and energy of modern laser systems long enough [1]. For this reason, the trend is toward active laser safety systems that can shut down the laser in a fraction of a second in the event of a hazard [2,3]. For this purpose, a sensor can be added to the passive protection material. If the laser beam destroys the opaque textile (Figure 1, right), the sensor sends a signal to an automatic switch-off device that immediately shuts down the laser. Some of these solutions are already available on the market, for example laser safety windows or laser safety walls. However, active systems for flexible and especially textile materials, such as laser safety curtains [3,4] or fabricated personal protective equipment (PPE) [5], are less common. The increasing use of mobile, high-power lasers, however, increases the demand for such textiles for active laser protection, since they can be used very flexibly, easily, and close to the process as laser protection. Thin-film solar cells based on hydrated amorphous silicon (a-Si:H) [6] can be used as sensors on laser protection textiles and fabrics. Thin-film coatings on flexible substrates for solar cells [7,8,9] could be much cheaper than on solid substrates using mass production trough roll-to-roll technology [10,11], as demonstrated by the companies Uni-Solar (Auburn Hills, MN, USA) and Hyet Solar (Arnhem, The Netherlands). On DuPont™ Kapton^®^ HN (Dupont, Wilmington, DE, USA), an energy conversion efficiency of 11.2% has been achieved for a-Si:H/nanocrystalline-Si:H in a n-i-p substrate configuration by [12] and 8.7% for a-Si:H cells on polyethylene naphthalate (PEN) with low temperature processes [13,14]. The active laser safety curtain of the company Jutec GmbH (Rastede, Germany) is currently certified for a wavelength range of 900–1080 nm [3]. Recently developed ultraviolet photodetectors with nickel- and copper-doped zinc oxide nanorods or NiO/ZnO thin films on flexible substrates could also be used for active laser protection applications in the future [15,16,17]. The presented active laser protection system with solar cells reacts to wavelengths of 250–810 nm and higher and leads to an extension of the active laser protection range. The focus here is on applications with high-power (cw—continuous wave) or high-energy (pulsed) lasers in the UV-VIS-NIR range.

## 2. Experimental Section

In order to cover as wide a range of wavelengths as possible for the active laser protection system, it was necessary to use a sensor that is sensitive in a spectral range from the ultraviolet to the near infrared. For this purpose, silicon solar cells seem to be well suited. However, they are mainly used to generate photocurrent and photovoltage through the solar spectrum and are continuously irradiated. In the following, the laser protection fabric and the solar sensor were investigated individually and then assembled in pulsed and continuous irradiation modes and with different laser wavelengths.

The laser safety textile consists of several layers as shown in Figure 1. On the outer sides there are the laser protection fabrics with a cover layer followed by a protective layer for a closed surface for the sensor on one side and an aluminum layer as threshold limit for the laser irradiation on the other side. In the inner part, the solar cell is located on a polymer foil. The laser protection materials consist of a polyester (PES) fabric with a basis weight of 600 g/m². The different used 10 fabrics were supplied by Protect Laserschutz GmbH (Nuremberg, Germany) with a yarn count of 204 tex. The main difference between the fabrics is the type of weave [18,19]. Plain weave (fabric 1,3,4,5,8), satin weave (fabric 2,7,9,10), and composite weave checkerboard (fabric 6) samples were provided (Figure 2). Within these weave types, the samples differed by weave variations; their coatings; and application temperatures, which were 200 °C and 500 °C. All fabrics have been tested and certified in accordance with the German DIN EN 12254 standard [20]. Vacuum and temperature stability tests were executed on a laboratory vacuum chamber system with pressure sensors (PR 35, Leybold, Cologne, Germany) and a tube furnace (R07/50, Heraeus, Hanau, Germany). For optical examinations a Zeiss Axio Imager.M2m microscope was used.

An experimental setup was developed for the laser characterization of the laser safety fabrics and the solar cells. There the substrate could be characterized with different lasers (see below) and variable configurations as shown in Figure 3. Each laser beam has been shaped with optical lenses to achieve similar geometric parameters. This allows measurements under the conditions of the German DIN EN 12254. The 1/e^2^ beam diameter of each laser was 1 mm except for the diode laser. The following lasers were used: an excimer laser (Compex 150T, Coherent, Gottingen, Germany) with a wavelength of 248 nm, a maximum energy density of 370 kJ/m² per pulse, and a pulse width of 20 ns; a cw Nd:YVO_4_ laser (Verdi 10, Coherent) with a wavelength of 532 nm and a maximum power density of 11.5 MW/m²; a Nd:YAG (503-D.NS 779/10, BMI) laser with a wavelength of 355 nm (third harmonic) and a maximum energy density of 95.5 kJ/m² per pulse, and a wavelength of 532 nm (second harmonic) with a maximal energy density of 102 kJ/m² per pulse and a pulse duration of 8 ns. The energy density values per pulse refer to a repetition rate of 10 Hz. The maximum power and energy density values are measured after the laser beam shaping. For measurement in the infra-red range a diode laser (900-L30 × 0.069-DL808-EX1010, LIMO) in cw-mode was used. The laser emits a wavelength of 808 nm and a maximal power density of 13.8 MW/m² with a 1/e² laser beam line profile of 30.6 × 0.21 mm² on the sample. A photodiode was used to synchronize the laser pulse with the measured voltage and current signals. A trigger signal is sent to the oscilloscope after the back reflection of the lens hits the diode detector. Using a beam splitter, the first power/energy meter measures the set laser power/energy. The laser beam is then directed at the sample. The second power/energy meter then measures the transmitted laser power/energy when the sample is damaged or destroyed.

For the solar cells, DuPont™ Kapton^®^ HN with a thickness of 0.127 mm or Rosco™ GamColor clear polyester foil with a thickness of 0.125 mm and Borofloat 33 glass with a thickness of 1.1 mm were used as references. After a cleaning step with acetone and isopropanol, the substrate was deposited with a 200-nm or 500-nm thick aluminum-doped zinc oxide (AZO) layer as front contact via atomic layer deposition (ALD) at a temperature of 225 °C. In the next step the hydrogenated amorphous silicon (a-Si:H) solar cell was deposited using plasma enhanced chemical vapor deposition (PECVD). The layer system consisted of a 10-nm highly p-doped a-Si:H emitter followed by an intrinsic amorphous silicon layer of 300 nm or 500 nm and a highly n-doped layer of 10 nm. The a-Si:H layers were deposited with a chamber pressure of 1.0 mbar; a heater temperature of 180 °C or 225 °C; a power of 4 W; and a gas flow of 1.5 sccm silane (SiH_4_), 1.45 sccm diborane (B_2_H_2_—for the p^+^ layer), and 1.45 sccm phosphine (PH_3_—for the n^+^ layer). As a last step a 200-nm thick silver contact layer was deposited using electron beam–physical vapor deposition (EB–PVD). For measurements on textiles the silver contact was replaced by an indium tin oxide (ITO) contact. The 200-nm ITO layer was deposited via DC magnetron sputtering with a basic pressure of 1 × 10^−5^ mbar and a work pressure of 5 × 10^−3^ mbar. The deposition rate was 0.18 nm/s at a power of 70 W. After deposition the layer was annealed at a temperature of 180 °C for 60 min. The performances of the solar cells were investigated using J–V measurements with a solar simulator (SS-80A, PET Inc., Ventura, CA, USA) under irradiation of 1000 W/m² with the air mass 1.5 global spectra (AM 1.5 g) and a Keithley 238 source measure unit. Next, the fabric was bonded to the Kapton film in a lamination process. The solar cell was then deposited directly onto the laser safety compound. Finally, a simple automatic switch-off device verified the active laser protection system for the different wavelengths and operating modes of the lasers.

## 3. Results and Discussions

### 3.1. Active Safety Concept

The widely used passive laser protection systems are made of materials that can withstand certain laser powers or energies for at least 100 s, depending on the protection level. For active laser protection, a sensor is added to these materials that switches off the laser within a certain time after the laser light is detected. In addition, other warning signals such as an acoustic signal or a shutter can be activated. In the system presented here, an a-Si:H solar cell is used as a sensor. The shutdown is realized by a simple automatic system with a limit switch. First, the most important individual components, the laser protection fabrics and the solar cells, were examined for their suitability for the planned application. Then, the laser protection composite system presented in Section 4 was investigated with special regard to the properties of the solar cells and laser resistance.

### 3.2. Laser Protection Fabric

The laser safety textile consists of several fabric components and an a-Si:H solar cell system that acts as a sensor for signal acquisition, as shown on the right in Figure 1. First, the laser protection fabrics were pre-selected for their solar cell suitability and laser resistance. For the latter, the irradiation tests were carried out agreeing to the standard for shielding at the workplace according to the German DIN EN 12254 with 100 s of irradiation time. The purpose of the test was to determine the power or energy density at which there would be destruction of the laser protection fabric. The level at which the laser irradiation begins to destroy the fabric was recorded using a power/energy meter behind the sample (Figure 4). Figure 4a shows the measured laser energy and power density values for different fabrics without damage or destruction. The respective protection levels can be determined on the basis of DIN EN 12254. The passive protection levels can be used to set threshold values for the active laser protection. Figure 4b displays the behavior during the damaging and eventual destruction of the protection fabric at a wavelength of 355 nm (blue in Figure 4a) and laser energy density of 25.5 kJ/m². For clarity, only 5 of the 10 fabrics are shown here. Measurements with a wavelength of 248 nm are shown in Appendix A. During irradiation with the green (532 nm) pulsed laser, no fabric destruction occurred up to the maximum laser energy density of 102 kJ/m².

### 3.3. Solar Cells

In the next step, the laser protection fabrics were tested for vacuum and temperature stability. For this purpose, the fabrics were placed in a vacuum system and the time until a pressure of 1 × 10^−5^ mbar was reached was measured. Only one fabric showed an outgassing behavior. For the temperature stability, a tube furnace was used. The samples were exposed to temperatures of 180 °C and 230 °C for 30 min. Here, the transparency and flexibility after exposure to temperature were evaluated. After testing of the laser protection, fabrics were deemed suitable for the deposition of the solar cell system. For the solar cells, a closed surface was needed. Therefore, the same procedures and tests as those undertaken for the fabrics were carried out with different coatings and foils, and in addition they were tested for their laser stability. Both Rosco™ GamColor clear polyester foil and DuPont™ Kapton^®^ HN met the criteria. However, subsequent measurements showed that the solar cells on the Rosco^TM^ polyester foil had lower reproducibility compared to Kapton, and the latter was then used for further experiments.

The structural design of a solar cell is shown in Figure 5. Here, a 1” substrate with a contact scheme with areas up to 28 mm² can be seen. This scheme was used mostly for the following laser irradiations. For upscaling, the substrate size was moved to 2” and larger.

Figure 6a shows the J–V curves for solar cells on Kapton and glass for reference. The J–V curve shows the solar cell diode characteristic with the current generated by light. The open-circuit voltage (V_oc_) is the highest possible voltage at zero current. The short-circuit current density (J_sc_) is the current through the solar cell at a voltage of zero related to the area of the solar cell. The fill factor is the ratio between the maximum electrical power and the product of V_oc_ and J_sc_ and the efficiency of the solar cell is the ratio of the maximum electrical power and the input power, here standardized to 1000 W/m² [21,22,23,24]. The measurements were made in the superstrate configuration; thus, the irradiation took place through the foil or the glass. In comparison, both samples have almost the same voltage. The Kapton sample shows a much lower short-circuit current value, which is due to the lower transparency of the foil and the transfer to the foil itself [7,14,25,26,27]. In the latter case, different expansion coefficients, outgassing during the depositions, and the gas permeation of the foil can influence the J–V parameters [14]. When increasing the solar cell area on glass samples, the voltage remains the same, but the current values decrease significantly due to insufficient contact (Figure 6b). Figure 6c,d demonstrates the reproducibility of the solar cells on glass and Kapton foil samples. All measured solar cells show photovoltaic activity, and the efficiencies still show potential for further improvements.

With further upscaling to solar cell areas of 16 cm² on glass, values of 900 mV and 4.5 mA/cm² could be achieved. When transferred to Kapton, increased short circuits of the solar cells occurred during initial tests. To prevent this, the film thickness of the AZO layer was increased to 500 nm and that of the a-Si:H layer system to 520 nm. After this, adjustments values of 890–900 mV on the Kapton foil could be measured.

Following the determination of the solar cell parameters, the sensors were irradiated with continuous wave and pulsed lasers in configuration as shown in Figure 3. It should be noted that the subsequent measurements of the voltage and current signals are not comparable with the V_oc_ and J_sc_ values. Since the laser energies and powers used correspond to more than 100 times the solar irradiance of 1000 W/m², the charge carriers of the solar cells are in saturation, which leads to a non-typical diode behavior [28,29,30]. A further point to consider is the inhomogeneous irradiation of the solar cell. Only a very small part (laser beam profile) of the cell area was irradiated with very high intensity. In this case, the equivalent circuit for an illuminated solar cell has to be extended by the equivalent circuit for a non-illuminated solar cell [31,32]. This circumstance will prove to be an advantage in further development. The displayed signals with the pulsed lasers do not correspond to the actual signals either. In order to determine the latter, a measurement setup was set up in a series of experiments. The electronic properties of the solar cell sensor were measured with a time resolution in the µs range, thus facilitating acquiring the signal at a lower sampling rate. This is more advantageous for the later switch-off electronics, and in the end, it is less expensive. The signal for 248 nm is also limited to the transmission of the 1.1-mm thick glass substrate [33]. Additionally, the shorter pulse duration for wavelengths of 355 and 532 nm leads to a different energy entry per pulse for the same energy level. The different decay times can be related to the different pulse widths of the lasers.

The signal curves of the solar cells at irradiation with the different lasers are presented in Figure 7. The behavior when irradiated without damage is shown in Figure 7a,c. It is clear from both diagrams that the sensor is detecting the laser radiation. Solar cell damage during laser irradiation is shown in Figure 7b,d. For the wavelength of 532 nm in the cw mode (Figure 6b), the signal values decrease significantly after damaging at a laser power density of 3.2 MW/m². Figure 6d shows the behavior of the solar cell in the pulsed laser mode. No significant changes were observed. Measurements with 355 nm show similar behavior (shown in Appendix A). Despite all of this, the solar cells indicate activity after damaging or localized destruction due to the scattered light, even with irradiation of the damaged area and its surroundings. This fact is very beneficial for the basic safety function of the solar cell sensor, even during and after serious damage.

Comparable measurements with solar cells on Kapton were performed. Although Kapton only transmits above a wavelength of 480 nm (below 480 nm the transmission is less than 1%), voltage and current signals induced by the high-power lasers can also be measured using the solar cell sensor.

Irradiations at a wavelength of 808 nm were also performed. Here, signals of several 100 mV could be detected even at low power densities of 1.5 kW/m². However, the signals show a rapid decrease as the silicon starts to crystallize (Appendix A). When the already damaged area was irradiated again, voltage signals could also be measured, albeit with lower signal strength. Since the beam profile of the laser is difficult to compare with the other lasers used, irradiation was only performed on the solar cells to show that the sensor is also sensitive for this wavelength.

### 3.4. Laser Protection Compound System

#### 3.4.1. Solar Cells on Laser Protection Fabric

The a-Si:H solar cells were directly deposited on the laser protection compound with the protection fabric—optionally with and without the aluminum foil—and a protection foil. The front contact (AZO layer) is located on the fabric side. So, the measurements for the solar cells were made with the silver contact layer on top, in a substrate configuration, as shown in Figure 8b. Due to the opaque silver the larger part of the solar cell area is covered. In order to be able to measure V_oc_ and J_sc_, the samples were arranged at an angle of 30° so that only reflections and scattered light contributed to the measurement. The 30° angle was determined empirically by rotating the sample under the sun simulator. At this position the highest voltage and current values were measured. This configuration was used as the new standard or reference for further function testing. For initial tests and verification of the solar cell function on the laser protection fabric, the silver contact was replaced by a transparent indium tin oxide (ITO) contact (Figure 9c) for a few samples [34]. In Figure 8a, solar cells on different substrate materials and laser protection fabric without the aluminum foil are compared. All measured samples show similar values for the current density and the voltage. Further increasing the solar areas up to 16 cm² resulted still in voltages of 28–140 mV. Therefore, the solar cells could be characterized for further experiments. Some bending tests were also performed with a low repetition number of 20× bending. Due to the rather inflexible laser protection compound, a radius up to 1.7 cm was used. The V_oc_ and J_sc_ values of the 1-cm² solar cells showed no changes before and after bending. To further optimize the bending behavior, silver nanowires can be added to the AZO layer [35].

Additional optimization could be achieved with thin aluminum oxide layers (Al_2_O_3_) as tunnel barriers between a-Si and the contact layer [36,37]. For the sensor concept presented here, these optimizations are not necessary, as the values achieved are sufficient for signal acquisition. However, this could become more important for future applications, like, e.g., for backpacks with plug-in mini solar modules directly deposited on the fabric. In addition, interconnected concepts with smaller areas of solar cells are also conceivable for personal protective equipment or other applications that use solar batteries and solar cells. Pretests showed a voltage of 6.2 V with an interconnection of 8 × 1 cm² solar cells in contrast to the one solar cell of 16 cm² with 900 mV. For the follow-up of this subject, there are now a large number of different sensor circuit concepts for smart textiles on the market [38].

#### 3.4.2. Validation of the Compound System

In the last development step, the behavior of the solar cells after damage of the fabric was examined. Initial experiments showed different light transmittance for the laser protection fabric. Thus, the solar cell sensors already detected scattered light from the lasers (Appendix A). So, there were two directions that could be followed. In the first one, the laser protection fabric was irradiated without the aluminum foil between the fabric and the solar cells, and the laser was switched off immediately. This effect can be used in laser-protection measures, e.g., when adjusting lasers. The other direction is to use thin aluminum foil. The foil can be applied as a threshold for a certain laser power or energy level. Only after destruction does the laser light reach the solar cell and the laser is switched off or an alarm signal triggered. In Figure 9 the measurements for solar cells on fabric with and without aluminum foil are presented. Figure 9a displays the voltage signal of the solar cell for irradiation at 248 nm and a 10 Hz repetition rate. The prolonged decay times of the pulses after fabric damage indicate increased scattered light around the damaged area. The same measurements were performed with a wavelength of 355 nm in the pulsed mode at 10 Hz. An increased decay time of the signals after damaging was also observed. Figure 9b) shows the irradiation of the solar cell on fabrics and a 10-µm thick aluminum foil in the cw mode. After destruction of the opaque laser protection compound, the measured current signal increased significantly. In both laser modes, the solar cell is very sensitive to the laser light. To test the function of the solar sensors, a simple automatic switch-off system for a high and a low mode was implemented. The system consists of a limit switch with a toggle switch for voltage or current setting and a two-way relay for shutter control or laser shutoff. This was used to test the irradiation of the fabrics with aluminum foil in the high mode and without aluminum foil in the low mode. When the signal increased, a shutter was triggered and/or an acoustic signal was generated at the same time. The limit switch was used to set the signal level at which the active protection system should switch off. A shutdown time of 50 ms was demonstrated successfully (Appendix A). An additional safety circuit is provided for the automatic switch-off system, which checks the sensor at intervals of a few 100 ms. When using laser protection fabrics with aluminum foil, the threshold value can be set using different thicknesses of the foil. Other opaque materials could also be used to extend the protection level. To protect the sensor, additional attenuators or filters can be placed in front of the solar cells.

The challenge for the active laser protection system was to find a textile or textile composite on which the solar cells could be deposited and which would still be active at the end of the process. The above measurements and results show that a-Si:H solar cells can serve as a sensor for the active switching of laser protection devices. The sensor is very sensitive to the laser wavelengths used, both in the pulsed mode as well as in the cw mode. With the developed automatic switch-off system, the active laser protection system was successfully tested.

## 4. Conclusions

In this manuscript, an active laser protection fabric system with a-Si:H solar cells as a large-scale sensor was presented. The fabrics and the solar cells were separately examined for their behavior during irradiation and damage at wavelengths of 245, 355, 532, and 808 nm in the continuous wave and pulsed modes. In the next phase the solar cell sensors were directly deposited on the laser protection fabric compound, and the damage and destruction behavior of the active laser protection system was investigated. The results showed that the safety function of the solar cell is maintained even if the previously destroyed area is irradiated again. With a simple automatic switch-off system, the active laser protection fabric reached a shutdown time of 50 ms. For laser protection textiles with or without aluminum foil, a low- or high-energy shutdown level could be established.

The laser safety system could be modularized by replacing the silver contact with an ITO contact, so that only the passive laser protection curtain needs to be replaced in case of damage. In addition, the use of glass fiber has shown promising results for solar cells in initial trials.

## Figures and Tables

**Figure 1 materials-16-04841-f001:**
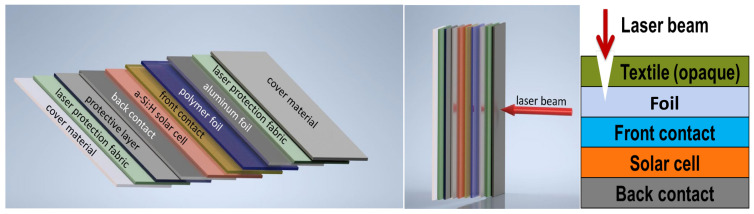
Layer structure of the laser safety textile.

**Figure 2 materials-16-04841-f002:**
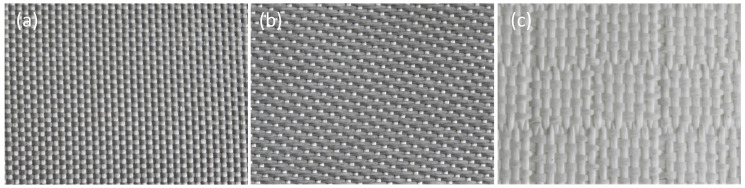
Examples of different weave types of the used laser protection fabrics; (**a**) plain weave, (**b**) satin weave, and (**c**) composite weave checkered.

**Figure 3 materials-16-04841-f003:**
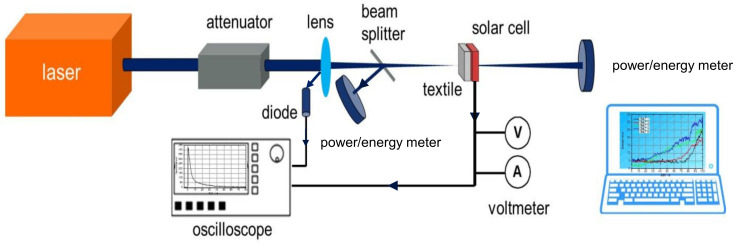
Scheme of the experimental setup for measurements of the laser protection fabric and solar cells.

**Figure 4 materials-16-04841-f004:**
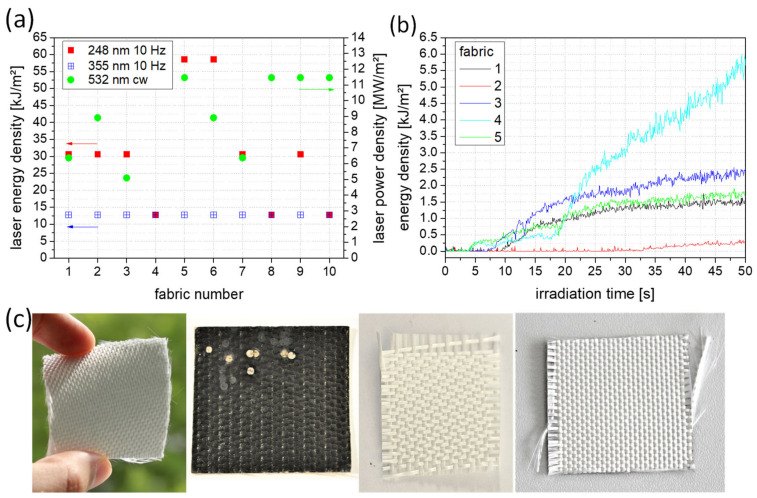
Behavior of different fabrics (**a**) during irradiation of certain energy and power density levels for an irradiation time of 100 s without destroying the laser protection fabric and (**b**) during destruction when irradiated with a wavelength of 355 nm and energy density of 25.5 kJ/m²; (**c**) pictures of different laser protection fabrics (from left to right: fabric 7, 1, 9, 3).

**Figure 5 materials-16-04841-f005:**
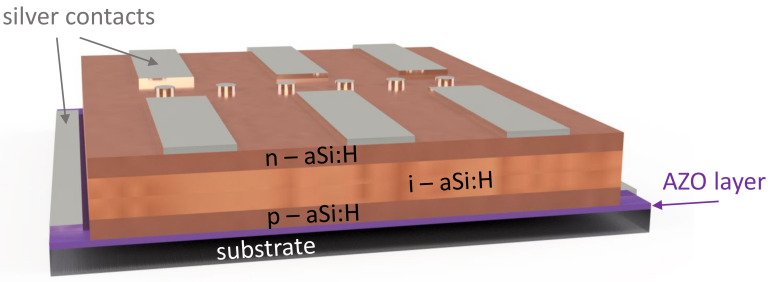
Schematic example of solar cells with silver contact on top with 6 × 28 mm² area (rectangular) and 6 reference cells with 0.8 mm² area (small circles).

**Figure 6 materials-16-04841-f006:**
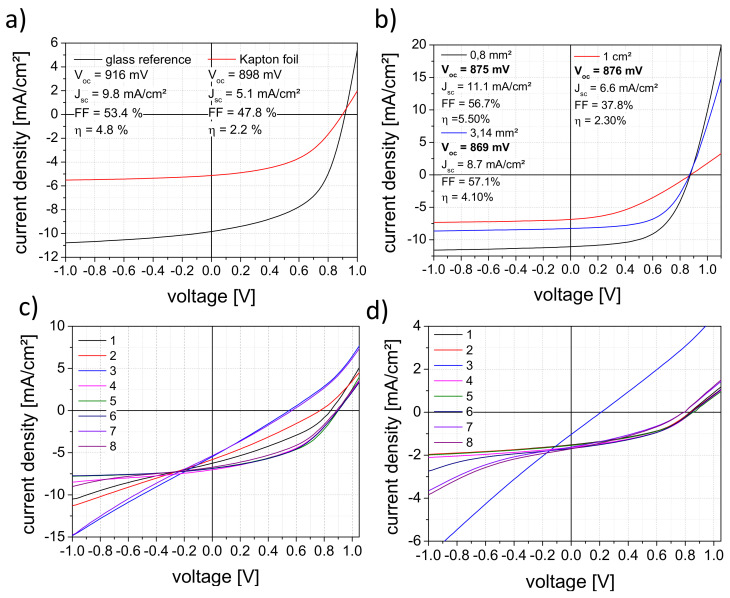
J–V parameters of a-Si:H solar cells on (**a**) glass and Kapton foil with an area of 0.8 mm², (**b**) on glass with upscaling behavior on different area sizes, (**c**) on 2” glass with a 8 × 1 cm² area, and (**d**) on 2” Kapton foil with 8 × 1 cm² area.

**Figure 7 materials-16-04841-f007:**
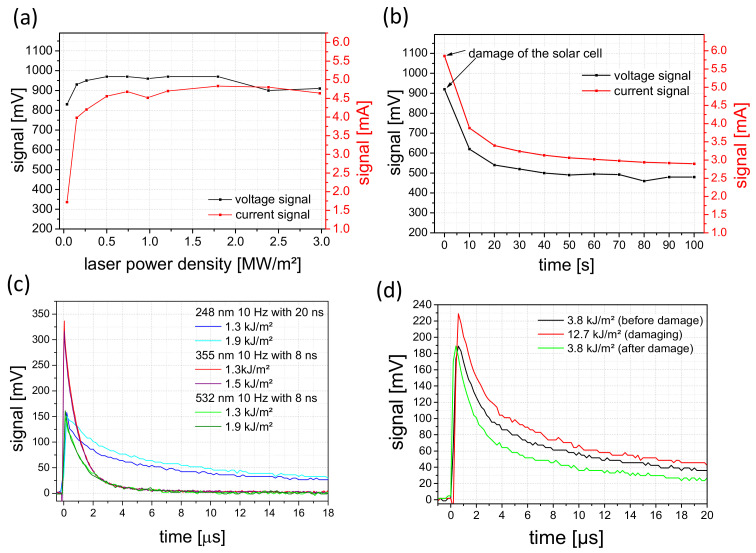
Signal curves of solar cells on glass at an irradiation (**a**) of 532 nm in cw mode without damaging and (**b**) with damaging the sensor. Part (**c**) shows voltage signals at different wavelengths and energy densities in pulsed mode without damaging, and (**d**) at a wavelength of 248 nm in pulse mode with damage to the solar cell sensor.

**Figure 8 materials-16-04841-f008:**
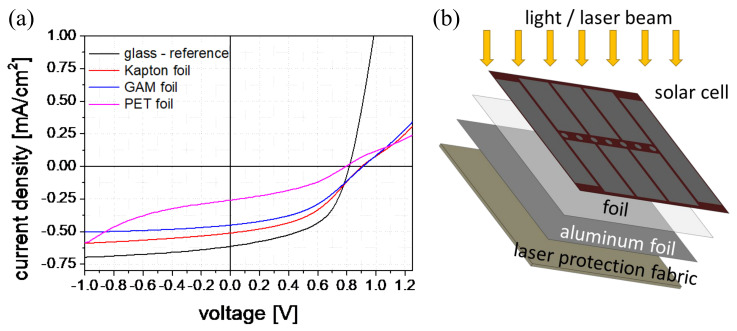
(**a**) J–V curves of solar cells on different substrates and laser protection fabric (PET—Polyethylene terephthalate) with areas of 28 mm², (**b**) schema of the fabric samples with the irradiation via silver contact with an angle of 30° in substrate configuration.

**Figure 9 materials-16-04841-f009:**
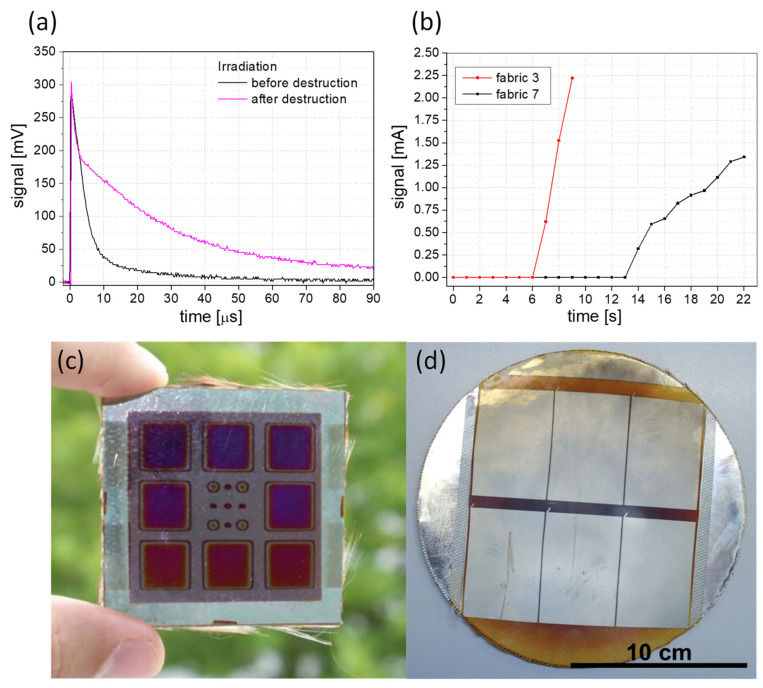
Irradiation signal of the solar cell on fabric for (**a**) a wavelength of 248 nm in pulsed mode with 10 Hz and 60 kJ/m² and (**b**) a wavelength of 532 nm in continuous wave mode with a power density of 10.2 MW/m² and aluminum foil with a thickness of 10 µm for two selected laser protection fabrics. Signal increase is interpreted as the beginning of irreversible damage. Solar cells (**c**) with ITO contact and 1-cm² areas and (**d**) with silver contact and 40-cm² areas directly deposited on laser protection textile with aluminum foil.

## Data Availability

The data presented in this study are available on request from the corresponding author.

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
