# Peer review of "Amorphous Silicon Thin-Film Solar Cells on Fabrics as Large-Scale Detectors for Textile Personal Protective Equipment in Active Laser Safetyâ€"

_materials, 2023, doi:10.3390/ma16134841_

Round 1
Reviewer 1 Report
very good research paper. In my opinion it can be published in its present form. I have two remarks/comments but it is not recessary to resubmit the paper for a re-reviewing.
1) line 120: laser resistance: is the laser light reflected, scattered or absorbed (and then converted into heat) in the top layer?
2) Are these solar cells wearable? If I walk in the sunshine with a shirt equipped with these solar cells, do they provide enough energy to charge my portable phone or any other wearable electronic equipment?
I repeat: I do not ask the authors to rewrite their paper. My comments are just useful information (I hope).
Congratulation with your nice work.
Reviewer 2 Report
The authors presented an active laser safety system based on amorphous silicon solar cells embedded onto laser fabric. A few issues need to be addressed:
1. As a solar cell on a fabric, one of the key desired properties from a Materials standpoint is the flexibility of the solar cells. The fabric is likely to be bent or even stretched many times over its lifespan. Hence, it would be important for the authors to test the solar cell durability upon bending.
2. The electronics and equipment for the automatic switch-off system needs to be described in detail.
3. Although the solar cells still "work" after being damaged by laser, I feel that the system could be more robust if the authors add a laser attenuator or OD filter, that still allows sensitive response by the solar cell without risk of damage by the laser.
Reviewer 3 Report
Dear Authors, please find my comments regarding the paper: Amorphous silicon thin film solar cells on fabrics as large-scale detectors for textile personal protective equipment in active laser safety.
Comment 1: "is still maintained at the edges when the previously destroyed area is irradiated again" - this is not understandable on its own when only reading the abstract.
C2: Figure is is not clear, how the consequential steps are following up, is there a feedback? Also, the basic scheme description is lacking. What is an active laser protection? What is the scientific literature behind it? The introduction needs serious improvements.
C3: Experimental section lacks the flow of the work, it describes the materials, but not the following methods in their logical sequences.
C4: It is not clear how F1-F10 fabrics are separated, and what are their main difference? If they are the same, how can the results be considered by statistical side?
C5: Figure 4 comes before actual discussion of the content. Fig 9 a -> "befor" -> "before".
C6: Please use vector exports and imports with the figures, as the bitmap basics are downscaled during PDF generation.
C7: I lack a final discussion before concluding the article. As you see, the main problem of the paper is the lack of cohesive flow. To see the motivation, to see how the hypothesis is achieved, and then when the results are done. how was the original hypothesis justified.
English is OK.
Reviewer 4 Report
The reviewer found the idea of the submitted manuscript titled ‘Amorphous silicon thin film solar cells on fabrics as large-scale detectors for textile personal protective equipment in active laser safety’ interesting and can be published in Materials. However, before finalizing this article for publication, please address the following points:
1. Authors have mentioned the diode in schematic 1, explain the explicit function of this device in the text.
2. In Fig 3, the authors mention the ‘Behavior of different fabrics’ while they present only one type of fabric picture.
3. Recent research articles citation is missing in the manuscript. Please include a few latest ones including the followings: in which inorganic material is used for UV-detectors and proposed that material for solar applications.
(a) Ultraviolet Photodetection Based on High-Performance Co-Plus-Ni Doped ZnO Nanorods
(b) High-performance flexible ultraviolet photodetectors with Ni/Cu-codoped ZnO nanorods grown on PET substrates
Round 2
Reviewer 2 Report
The manuscript has improved after revision, and I am willing to accept it.
Author Response
Thank you very much.
Reviewer 3 Report
Dear Authors:
C1. the scheme on Figure 3 is still not perceivable on its own. Please indicate directions. (From laser to meter it is clear.)
C2. i still feel that the experimental description lacks a part where the motivation/hypothesis is presented, then immediately the methods are listed, and a possible outcome is predicted. So that the reader, when follows up with the detailed experimental methodologies and results.
Round 3
Reviewer 3 Report
Thank you for the corrections.